# Tattoo-Associated Skin Reaction in a Melanoma Patient Receiving B-RAF and MEK Inhibitors: A Case Report with an Emphasis on Etiopathogenic and Histological Features

**DOI:** 10.3390/jcm13020321

**Published:** 2024-01-06

**Authors:** Silvia Baratta, Gerardo Cazzato, Caterina Foti, Giuseppe Ingravallo, Lucia Lospalluti, Carmelo Laface, Raffaele Filotico, Francesca Ambrogio

**Affiliations:** 1Section of Dermatology and Venereology, Department of Precision and Regenerative Medicine and Ionian Area (DiMePRe-J), University of Bari “Aldo Moro”, 70124 Bari, Italy; silviabaratta94@gmail.com (S.B.); caterina.foti@uniba.it (C.F.); l.lospalluti@gmail.com (L.L.); raffaele.filotico@uniba.it (R.F.); dottambrogiofrancesca@gmail.com (F.A.); 2Section of Molecular Pathology, Department of Precision and Regenerative Medicine and Ionian Area (DiMePRe-J), University of Bari “Aldo Moro”, 70124 Bari, Italy; giuseppe.ingravallo@uniba.it; 3Medical Oncology, Dario Camberlingo Hospital, 72021 Francavilla Fontana, Italy

**Keywords:** tattoos, immune checkpoint inhibitor, adverse reaction, granuloma, molecular targeted therapies, tattooing

## Abstract

Tattoo-associated cutaneous reactions have become quite frequent given the increasing percentage of tattooed subjects globally and also in Italy. On the other hand, the increasing use of target therapy is showing the ability of these drugs to affect the immune system and also cause adverse tattoo-related reactions. In this paper, we report a case of a 42-year-old patient with stage-IIID melanoma undergoing treatment with Dabrafenib and Trametinib. The patient reported erythema, oedema and scaling in areas of the body containing a black tattoo, and, conversely, no signs and/or symptoms in areas with tattoos of a different color. Histopathological and immunohistochemical features indicated a lympho-histiocytic reaction with a granulomatous morphology, mainly distributed around the vessels and hair adnexa. By discussing the cases reported in the literature prior to ours, we concluded and provided the possible indications of the pathogenesis.

## 1. Introduction

Tattoo-associated skin reactions are frequent manifestations since, in the last decades, tattooing has become a practice that has enormously increased in popularity [1]. Globally, the incidence of tattooing reaches 36% for under-40-year-old people [2]. In Italy, from the data of a survey carried out by the Istituto Superiore di Sanità (ISS) concluded in 2015, it is assumed that there are 6.9 million tattooed people, i.e., 12.8% of the population, a value that rises to 13.2% if ex-tattooed people are also considered [3]. A study conducted on a sample of 7608 people, representative of the Italian population aged 12 and over, showed that those who get tattooed do so mainly for a ‘hedonistic’ choice: tattoos as a decoration, ornamentation and embellishment of the body, but there is also 0.5% who get tattooed for medical purposes (e.g., tattooing of the areola–nipple complex or endoscopic tattooing) and 3% for aesthetic purposes (so-called permanent make-up). From the result of this study, tattoos are more common among women (13.8% of respondents) than men (11.7%). Furthermore, gender differences emerge with respect to the location of the tattoo, with men preferring to tattoo arms, shoulders and legs, and women mainly the back, feet and ankles [3].

The tattooing process involves the mechanical deposition of ink under the skin by means of a special mechanical instrument, operated by a foot pedal and equipped with one or more needles (sometimes up to 35 needles) that penetrate the skin 1/2 mm at a speed that can vary between 50 and 3000 strokes, cycles per minute [4]. Usually, tattoo inks contain organic pigments, but they also include preservatives and contaminants such as nickel, chromium, manganese or cobalt, and in addition to carbon black, another most commonly used ingredient in tattoo inks is titanium dioxide (TiO_2_), a white pigment usually applied to create certain shades when mixed with dyes. In any case, all inks are composed of pigments combined with a carrier [5]. Considering the above, it is rather intuitive to understand how the introduction of exogenous compounds may stimulate a local response of different kinds. Clinically, adverse cutaneous tattoo-related reactions are classified as acute inflammatory reactions, allergic reactions from hypersensitivity, pseudolymphoma-type and granulomatous reactions [6]. Most granulomatous reactions are observed in patients with sarcoidosis and, more rarely, are drug-induced [7,8]. Eruptions under tyrosine kinase inhibitors like Dabrafenib and Trametinib are an uncommon immune-related adverse event that can become more frequent with the increased use of this therapeutic class of drugs [8]. In the recent review by Kluger N., 10 cases (six men and four women) treated with a combination of BRAF inhibitor (BRAFi) and a MEK inhibitor (MEKi), and PD-1/PD-L1 inhibitor (PD1/PD-L1i) (2) and CTLA-4 inhibitor (CTLA4i), including 1 case in combination with PD-1i, respectively, were isolated [8].

In this paper we present a rare case of a 42-year-old man with an history of stage-IIID of melanoma, B-RAF-mutated, receiving treatment with target therapy Trametinib and Dabrafenib, that developed erythema and desquamation near the black ink of tattoos; furthermore, we discuss the case with parallelism and differences to other cases present in the literature.

## 2. Case Presentation

We present the case of a 42-year-old patient who has been treated with Dabrafenib (150 mg bid) and Trametinib (2 mg) for a BRAFV600E-mutated melanoma at stage IIID (according to the 8th Edition of the American Joint Committee on Cancer stage system) since July 2022. 

The patient came to our attention in January 2023 (after 7 months of therapy) for the appearance of erythema, oedema and significant flaking (Figure 1A,B) associated with itchy symptoms only for tattoos made with black pigment, saving the colored ones (Figure 2A,B). Erythema was noticeable, but even more pronounced were the oedema and scaling. No papules or infiltrated plaques on or around the tattoos were noted. No xerosis or scaling was evident on the rest of the skin area nor at the contralateral elbow. The scaling was probably evident as an outcome of the inflammatory process. Moreover, the patient reported that the aforementioned reactions were more pronounced in recent tattoos than in older ones.

Patch tests with the SIDAPA (Società Italiana di Dermatologia Allergologica Professionale e Ambientale) baseline series (Euromedical, Calolziocorte, Italy) were performed. Patch tests were applied on the back in occlusion for three days with an Al test (Euromedical) on the Scanpor Tape (Norgesplaster, Vennesla, Norway). Test evaluations were performed on days 3 (D3) and 7 (D7) and showed a negative reaction. 

The skin biopsy (punch, 6 mm) revealed a moderate lymph–histiocytic inflammatory infiltration in an abundant proportion of blackish exogenous pigment. The distribution of the infiltrate was granulomatous and more concentrated around the vessels and the adnexa (Figure 3A–D). In particular, there was a lot of laden-black-pigment macrophages (Figure 3C,D).

Immunohistochemical investigations showed the diffuse presence of macrophages highlighted by CD68 (PG-M1, Dako, 1:800 dilution) and CD163 (Polyclonal, ThermoFisher 1:50 dilution) with exogenous black pigment (tattoo) inside the cytoplasm (Figure 4A,B) and distributed around the vessels and adnexa, always in a granulomatous pattern. Furthermore, immunoreactions for CD3 lymphocytes showed many cells together within the histiocytic component (Figure 4C). There were few B cells highlighted by the CD20 antibody (Figure 4D).

High-efficacy topical corticosteroid was prescribed, with the complete resolution of cutaneous manifestations within two weeks of treatment. The discontinuation of systemic treatment with Dabrafenib and Trametinib was not necessary.

## 3. Discussion

Dating back to ancient times, the practice of tattooing has always been used for different social purposes, mostly symbolizing groups, ethnicity, location, love, profession and religion. The etiology of tattoo-associated skin reactions is not easy to define, since clinical and histological data are usually non-specific. Indeed, even in tattoo-related allergic reactions, the most frequent ones, patch tests can often be negative. This is mainly because the pro-allergic hapten may not be present in the tattoo ink, but it can be a by-product formed de novo in the dermis after metabolization or via photodegradation [9]. 

As reported by various reports in the literature, the most common pigments that are responsible for allergic reactions are red and black [10]. Even in a brief review on tattoo-associated skin reactions in patients treated with targeted therapies and immune checkpoint inhibitors for advanced cancers, 86% of reactions occurred in dark/black tattoos [8], like in our case report.

Skin biopsy is fundamental to study any delayed or persistent reaction in the tattooed area, and histological examination may rule out mycobacterial infections, systemic diseases or lymphoma infiltration [10]. It is important to underline that tattoo-associated allergic reactions may not have typical histopathological features. Indeed, they may also occur without the pattern of spongiotic dermatitis because the allergen is inoculated directly into the dermis. The most suggestive histological picture of tattoo-associated allergic reaction is the prevalence of lymphocytes in the inflammatory infiltrate, with or without eosinophils. This ‘band infiltrate’ is found in the superficial dermis and also consists of mononucleate cells and macrophages loaded with pigment. More rarely, allergic reactions to tattoos may occur with granulomatous or pseudo-lymphomatous histological patterns (B- and T-cell types). In the presence of a granulomatous pattern, granulomatous diseases and microbial or fungal infections have to be excluded. In pseudo-lymphoma suspicion, however, polyclonality of the inflammatory infiltrate should be demonstrated via molecular investigations [11,12,13,14]. 

In this regard, it is important to mention a 2021 study [15] conducted at a tertiary care center in India on a cohort of 22 patients. In this manuscript, the authors, starting from a population of 1963 patients, analyzed the 1.1% (n = 22) who had presented different types of skin reactions following a previous tattoo. With an 18-month prospective observational design, the data of these 22 patients aged between 17 and 35 years (average age 24.6 years) were analyzed. The authors observed (as previously highlighted) that black/dark ink was more responsible for skin reactions (15 patients) and there was a greater possibility of adverse outcomes when the tattooing was carried out by amateurs rather than professionals.

From a clinical point of view, the skin manifestations were evident from a minimum of 3 months to a maximum of 24 months, with an average of 8.1 months, and consisted of the itching of hyperpigmented areas, mildly eczematous plaque developing over the tattoos or erythematous pruritic plaques in a few patients.

Histopathologically, hyperkeratosis was one of the most frequent findings (observed in 14 patients) followed by parakeratosis (10), spongiosis (9), acanthosis (5) and pseudo-epithelial hyperplasia (3). Dermal changes showed chronic inflammatory infiltrate comprising mainly lymphocytes (8), neutrophils (4) and histiocytes (4), mainly distributed in the perivascular area, such as in our case. Chronic inflammatory infiltrate along the dermo-epidermal junction was seen in three patients, and ten patients showed the presence of pigment in various layers of the dermis.

Drug-induced tattoo reactions are not very frequent AEs, although tattoo-associated skin reactions have been reported during immune restoration syndrome with highly active antiretroviral therapy for human immunodeficiency virus [16], therapy for hepatitis C [17,18] and tumor necrosis factor α inhibitors [19]. Immune checkpoint inhibitors are also associated with various dermatologic manifestations, including sarcoidosis and granulomatous reactions to tattoos [8,20].

To the best of our knowledge, only six cases of skin reaction to tattoos during BRAF and MEK inhibitor therapy, like with our patient, have been reported in the literature: three of them describe directly granulomatous findings at biopsy [21,22,23] and the other ones only mild lymphocytic infiltrations [24,25,26]. There are no cases reported in the literature of skin reactions to tattooing during alternative target therapies for melanoma such as Vemurafenib and Cobimetinib. However, the phenomenon is probably underdiagnosed. Our case is the only one in the literature in which the presence of clinical manifestations is reported only for black tattoos, sparing those with pigments of other colors. Also, the one in which the reaction is more pronounced was among the newer tattoos, probably because of the greater amount of pigment in these. Is there a greater reaction to new tattoos because more antigen is present in the macrophages? The trigger of these reactions is difficult to define because the molecular mechanisms of targeted therapy are complex. Tattoo ink applied using needles remains stored in macrophages or fibroblasts, and even migrates into the lymph nodes [27]. Macrophages are the most abundant inflammatory cells in melanomas [28]. The number of infiltrating macrophages and the levels of macrophage-produced factors inversely correlates with patients’ outcomes in both the early and late stages of melanoma [28]. Melanoma-associated macrophages produce many growth factors, cytokines, chemokines, extracellular matrix and proteinases, which play critical roles in melanoma initiation, angiogenesis, growth, metastasis and immune suppression [28]. However, the role of macrophages in melanoma resistance to BRAF inhibitors remains poorly defined. Moreover, BRAF inhibitors induce paradoxical activation of the MAPK pathway in macrophages, leading to profound effects on both macrophages and tumor cells through the production of VEGF with complex signal regulation [28]. These macrophages could release allergens presented in ink tattoos and trigger an allergic reaction.

The reaction could also be due to BRAF inhibitors that induce a reduction in myeloid-derived suppressor cells that can trigger allergic contact dermatitis [24]. 

Instead, a granulomatous reaction could be stimulated through the paradoxical activation of the ERK [27] or AKT/mTOR pathways [29]. 

There is still much to be clarified about the mechanisms underlying these reactions, and our case is intended to enrich the present literature on the topic in order to obtain more data about it. 

## 4. Conclusions

In this paper, we report a clinical case of a patient affected by stage-IIID BRAF-mutated melanoma that developed a tattoo-associated skin reaction under Dabrafenib and Trametinib therapy. The reaction was treated with a topic corticosteroid, with the complete resolution of the dermatological manifestation. The suspension of systemic treatment was not necessary, unlike in the other cases reported in the literature. However, other studies are needed to better explain the etiological mechanisms of this reaction to tattoos. 

On the basis of the reported date, patients with tattoos who need targeted therapy with anti-BRAF and anti-MEK should be informed about the risk of a tattoo-associated skin reaction, and a clinical examination of tattoos should be performed periodically.

## Figures and Tables

**Figure 1 jcm-13-00321-f001:**
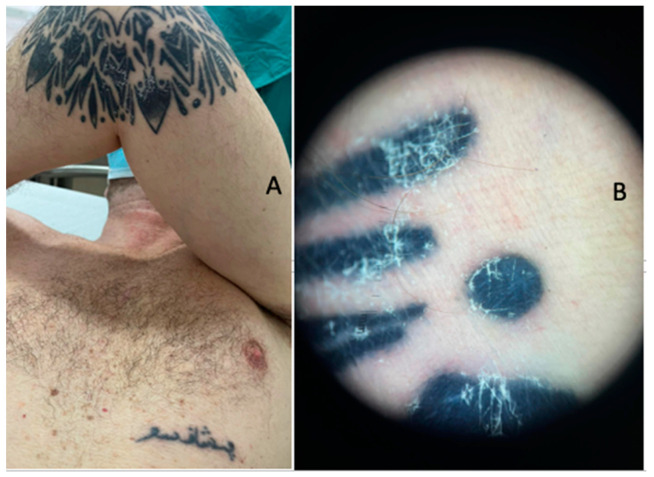
(**A**) Clinical examination of the left arm of the patient with the black-ink tattoo; the erythema, oedema and desquamation on the body area with the tattoo should be noted. (**B**) Dermoscopical examination: the erythema and desquamation near the area with the black ink should be noted.

**Figure 2 jcm-13-00321-f002:**
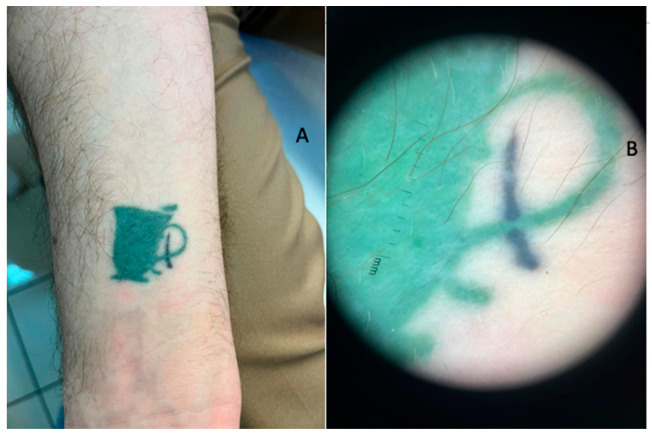
(**A**) Clinical examination of the right forearm with the green ink tattoo; the almost-absent inflammation in this area should be noted. (**B**) Dermoscopical features showing the previous clinical characteristics.

**Figure 3 jcm-13-00321-f003:**
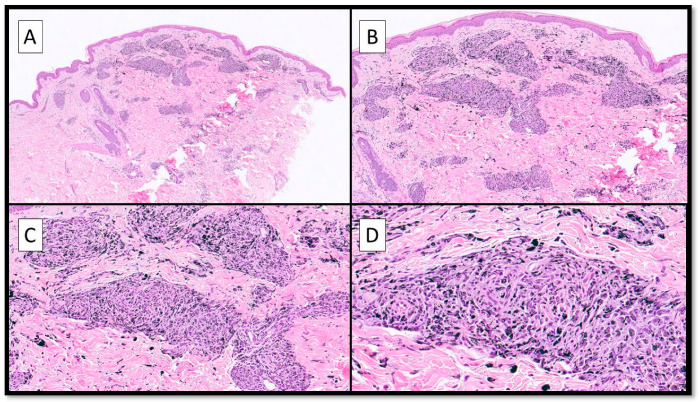
(**A**) Overview of the skin biopsy in the area of the black-ink tattoo; the pseudo-granulomatous distribution of the immune cells around the vessels and, sometimes, the adnexa (hematoxylin–eosin, Original Magnification 4×) should be noted. (**B**) Higher magnification of the previous picture showing the distribution of the lymphocytes and macrophages around the black exogenous pigment (hematoxylin–eosin, Original Magnification 10×). (**C**,**D**) Details of the previous histological picture showing lymphocytes and laden-pigment-macrophages around the vessels and adnexa (hematoxylin–eosin, Original Magnification 20× and 40×).

**Figure 4 jcm-13-00321-f004:**
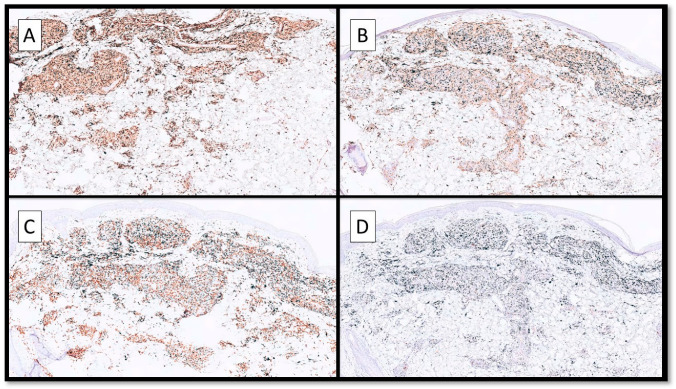
(**A**) Immunohistochemical investigation of CD68 (PG-M1) showing the abundance of macrophages in a granulomatous pattern with the black-ink pigment inside the cytoplasm; the perivascular and peri-adnexal distribution of the infiltrate (immunohistochemistry for CD68, Original Magnification 20×) should be noted. (**B**) Immunohistochemical photomicrograph for CD163 showing the same features of distribution and density of the cells seen in picture A with CD68; note that CD163 is a specific lineage histiocytic marker (immunohistochemistry for CD163, Original Magnification 10×). (**C**) Photomicrograph showing the distribution of CD3+ T cells (immunohistochemistry for CD3, Original Magnification 10×). (**D**) Photomicrograph showing CD20 B cells; the almost-complete negativity of the reaction (immunohistochemistry for CD20, Original Magnification 10×) should be noted.

## Data Availability

No new data were created or analyzed in this study. Data sharing is not applicable to this article..

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
