# Peer review of "Tattoo-Associated Skin Reaction in a Melanoma Patient Receiving B-RAF and MEK Inhibitors: A Case Report with an Emphasis on Etiopathogenic and Histological Features"

_jcm, 2024, doi:10.3390/jcm13020321_

Round 1

Reviewer 1 Report

Comments and Suggestions for Authors I read a manuscript by Silvia Baratta al, in which the authors describe the case of tattoos-associated skin reaction in a patient with Melanoma treated with B-RAF and MEK inhibitors. So far numerous articles confirm that treatment with B-RAF and MEK inhibitors is associated with several side effects, including those of the skin. The structure of the article is in my opinion very well designed, I appreciate the authors' contribution to the preparation of the article including the histopathological and immunohistochemical images. However, I have some fundamental comments on the content of the article itself. First, the Authors stated that  „Patient came to our attention for the appearance of erythema, oedema and flaking „.As the Authors refer to the article by Kluger N. (Tattoo Reactions Associated with Targeted Therapies and Immune Checkpoint Inhibitors for Advanced Cancers: A Brief Review. Dermatology. 2019;235(6):522-524. ) it should be emphasised that in all the cases described, the skin manifestations were significantly enhanced, ranging from erythema to even papules. In the reviewed article in the described patient there was no erythema and only desquamation was clearly visible. It may be related to increased dryness as one of the most common adverse reactions effects of the Dabrafenib treatment. Furthermore, the location of dryness and desquamation on the skin of the elbow is justified by the greater dryness of the skin in this area. Furthermore, authors do not mention what the skin of the other elbow looked like as well as do not mention the time of onset of the described skin reaction. The results of skin biopsy of the tattoo are in no way surprising, as macrophages are inherently involved in the removal of foreign bodies and are detectable in biopsy specimens of all tattoos- see article Høgsberg T, Thomsen BM, Serup J. Histopathology and immune histochemistry of red tattoo reactions. Interface dermatitis is the lead pathology, with increase in T-lymphocytes and Langerhans cells suggesting an allergic pathomechanism. Skin Res Technol. 2015;21(4):449-458. doi:10.1111/srt.12213  

Author Response

Comments and Suggestions for Authors

I read a manuscript by Silvia Baratta al, in which the authors describe the case of tattoos-associated skin reaction in a patient with Melanoma treated with B-RAF and MEK inhibitors. So far numerous articles confirm that treatment with B-RAF and MEK inhibitors is associated with several side effects, including those of the skin. The structure of the article is in my opinion very well designed, I appreciate the authors' contribution to the preparation of the article including the histopathological and immunohistochemical images. However, I have some fundamental comments on the content of the article itself. First, the Authors stated that “Patient came to our attention for the appearance of erythema, oedema and flaking” as the Authors refer to the article by Kluger N. (Tattoo Reactions Associated with Targeted Therapies and Immune Checkpoint Inhibitors for Advanced Cancers: A Brief Review. Dermatology. 2019;235(6):522-524.) it should be emphasised that in all the cases described, the skin manifestations were significantly enhanced, ranging from erythema to even papules. In the reviewed article in the described patient there was no erythema and only desquamation was clearly visible. It may be related to increased dryness as one of the most common adverse reactions effects of the Dabrafenib treatment. Furthermore, the location of dryness and desquamation on the skin of the elbow is justified by the greater dryness of the skin in this area. Furthermore, authors do not mention what the skin of the other elbow looked like as well as do not mention the time of onset of the described skin reaction. The results of skin biopsy of the tattoo are in no way surprising, as macrophages are inherently involved in the removal of foreign bodies and are detectable in biopsy specimens of all tattoos- see article Høgsberg T, Thomsen BM, Serup J. Histopathology and immune histochemistry of red tattoo reactions. Interface dermatitis is the lead pathology, with increase in T-lymphocytes and Langerhans cells suggesting an allergic pathomechanism. Skin Res Technol. 2015;21(4):449-458. doi:10.1111/srt.12213.

Re: Thank you for your suggestions.

Compared to the ones described by Kluger, in our case no papules were noted but the itchy symptoms were so pronounced that the patient had to be admitted to our ER. The erythema was noticeable but even more pronounced were the edema and scaling; to this point, no xerosis or scaling was evident in the rest of the skin area nor at the contralateral elbow: probably the scaling was evident as an outcome of the inflammatory process. The onset of clinical manifestations occurred about 7 months after the beginning of target-therapy with Dabrafenib and Trametinib, which started in July 2022 at a dosage of 150 MG BID and 2 MG/day, respectively.

Clinical and symptomatic remission was observed after about 15 days of therapy with clobetasol ointment BID.

Regarding the histopatology, we read, analyzed, cited and extensively discussed the paper suggested with important parallelism and differences with our case.

Reviewer 2 Report

Comments and Suggestions for Authors

The authors describe a case of a patient with tattoos receiving treatment for stage IIID melanoma with BRAF/MEK inhibitors. The patient developed a skin reaction in the areas where the tattoos were placed. 

The case description is robust. Perhaps the introduction can be truncated to maintain the reader's interest. The discussion section should also discuss other relevant adjuvant therapy options for patients with melanoma. 

Author Response

Comments and Suggestions for Authors

The authors describe a case of a patient with tattoos receiving treatment for stage IIID melanoma with BRAF/MEK inhibitors. The patient developed a skin reaction in the areas where the tattoos were placed.  

The case description is robust. Perhaps the introduction can be truncated to maintain the reader's interest. The discussion section should also discuss other relevant adjuvant therapy options for patients with melanoma. 

Re: Thank you.

As suggested, we have shortened the opening section. 

There are no cases reported in literature of skin reaction to tattooing during alternative target therapies for melanoma such as Vemurafenib + Cobimetinib.

Reviewer 3 Report

Comments and Suggestions for Authors

In the manuscript titled ‘Tattoos-associated skin reaction in a patient with Melanoma under B-RAF and MEK inhibitors therapy: a case report with emphasis on etiopathogenic and histological features’, the authors presented a case report of a Melanoma patient given certain chemotherapy and manifesting a skin reaction associated with inked tattoo. The authors further highlighted their findings in light of the relevant literature. The manuscript is sound and data is presented in a meticulous manner.

Comments:

Abstract

The second sentence is quite long and need to be shortened to make consise statements as the background of the study is highlighted here.

The exact conclusion of the study is not mentioned in the abstract.

Introduction

In line 64, authors mentioned the study as a review but the study contained participants, it should be clear what type of study is referenced.

The authors should mention a link between erythema and desquamation with the chemotherapy or components in the ink as the rationale for this study.

Case Presentation

The total duration of the treatment is missing for the case study.

The authors mentioned old and new tattoos. The complaint for newer tattoos, is it linked to treatment or the disease condition is hard to say, authors should elaborate on this issue.

The figures in terms of differences are not explained in the test, more explanation is required especially for figure 3 and 4.

Discussion

The discussion is well written, however authors should also focus on mentioning the lack of studies in addition to the said difficulties.

The questions mentioned in the last paragraph of the discussion should be made into bullet points and given citations and explanation for easy understanding of the link to the present study.

Conclusion

The conclusion of the study is broad and is okay because there is still a lot unknown.

Comments on the Quality of English Language

There are some minor grammatical errors and also sentence structure related flaws in the manuscript which require proof reading.

Author Response

Comments and Suggestions for Authors

In the manuscript titled ‘Tattoos-associated skin reaction in a patient with Melanoma under B-RAF and MEK inhibitors therapy: a case report with emphasis on etiopathogenic and histological features’, the authors presented a case report of a Melanoma patient given certain chemotherapy and manifesting a skin reaction associated with inked tattoo. The authors further highlighted their findings in light of the relevant literature. The manuscript is sound and data is presented in a meticulous manner.

Abstract

The second sentence is quite long and need to be shortened to make consise statements as the background of the study is highlighted here. The exact conclusion of the study is not mentioned in the abstract.

Introduction

In line 64, authors mentioned the study as a review but the study contained participants, it should be clear what type of study is referenced.

The authors should mention a link between erythema and desquamation with the chemotherapy or components in the ink as the rationale for this study.

Case Presentation

The total duration of the treatment is missing for the case study.

The authors mentioned old and new tattoos. The complaint for newer tattoos, is it linked to treatment or the disease condition is hard to say, authors should elaborate on this issue.

The figures in terms of differences are not explained in the test, more explanation is required especially for figure 3 and 4.

Discussion

The discussion is well written, however authors should also focus on mentioning the lack of studies in addition to the said difficulties.

The questions mentioned in the last paragraph of the discussion should be made into bullet points and given citations and explanation for easy understanding of the link to the present study.

Conclusion

The conclusion of the study is broad and is okay because there is still a lot unknown.

Comments on the Quality of English Language

There are some minor grammatical errors and also sentence structure related flaws in the manuscript which require proof reading

Re: Thank you very much for your suggestions.

Abstract

  • As suggested, we provided to shorten the second sentence of the abstract.
  • The conclusions of the case are not mentioned in the abstract since there are no sure hypotheses about the pathogenesis of this reaction: the mechanisms behind target therapy are not yet fully elucidated especially about the alterations that this therapy induces on the immune system of the patients treated.

Introduction

  • About Kluger N.’ review mentioned in line 64, the patients we referred about are just the ones reported in the review of literature made by the author as an actual statement about tattoos-associated adverse reactions during immunotherapy and target therapy. We have added the refenference.
  • About the clinical skin manifestations, we suppose that the erythema is consequent to the inflammatory process (confirmed by skin biopsy) and that the scaling, present only at the level of the tattooed areas and not over the whole skin area, is also a consequence of the inflammation.

Case presentation

  • In line 80, we added the time period the patient came to our attention to define the total duration of treatment at the time of presentation of the clinical manifestations.
  • We assume that the reaction is more pronounced on the newer tattoos since the amount of pigment is greater in these.
  • 3 e 4 Gerardo

Discussion

  • This case is peculiar because it joins a quite poor literature on the topic and helps to support the hypothesis that the target therapy is able to cause immune alterations inducing various clinical manifestations. Another peculiarity is that our case is the only one in literature in which the presence of clinical manifestations is reported only on black tattoos sparing those with pigments of other colors.
  • As a result of the comments and suggestions obtained after the review, we would have thought of removing the part containing the questions and focus on the peculiarity of our case being the third currently present in literature of tattoo reaction during target therapy in which there is a granulomatous picture at histology

Conclusions

Thanks for your comment.

Reviewer 4 Report

Comments and Suggestions for Authors

This is a nice and well-documented case report on tattoos-associated skin reaction in a patient with melanoma under B-RAF and MEK inhibitors therapy. However, it is neither original nor brings new insight into the pathogenesis of this reaction. The introduction regarding the tattooing process seems a bit long.

Comments on the Quality of English Language

Minor editing of English language required

Author Response

Comments and Suggestions for Authors

This is a nice and well-documented case report on tattoos-associated skin reaction in a patient with melanoma under B-RAF and MEK inhibitors therapy. However, it is neither original nor brings new insight into the pathogenesis of this reaction. The introduction regarding the tattooing process seems a bit long.

Re: Thank you for your comment.

  • This case is peculiar because it joins a quite poor literature on the topic and helps to support the hypothesis that the target therapy is able to cause immune alterations inducing various clinical manifestations. Another peculiarity is that our case is the only one in literature in which the presence of clinical manifestations is reported only on black tattoos sparing those with pigments of other colors and also the one in which the reaction is more pronounced on the newer tattoos probably because of the greater amount of pigment in these.

Furthermore, it is the third case currently present in literature of tattoo reaction during target therapy in which there is a clear granulomatous picture at histology.

Round 2

Reviewer 1 Report

Comments and Suggestions for Authors

As modified, I accept the manuscript in this form